# Urinary Protein Array Analysis to Identify Key Inflammatory Markers in Children with IgA Vasculitis Nephritis

**DOI:** 10.3390/children9050622

**Published:** 2022-04-27

**Authors:** Julien Marro, Andrew J. Chetwynd, Rachael D. Wright, Silothabo Dliso, Louise Oni

**Affiliations:** 1Department of Women’s and Children’s Health, Institute of Life Course and Medical Sciences, University of Liverpool, Liverpool L12 2AP, UK; j.g.marro@liverpool.ac.uk (J.M.); a.j.chetwynd@liverpool.ac.uk (A.J.C.); rachael.wright@liverpool.ac.uk (R.D.W.); 2NIHR Alder Hey Clinical Research Facility, Clinical Research Division, Alder Hey Children’s NHS Foundation Trust, Liverpool L14 5AB, UK; silothabo.dliso@alderhey.nhs.uk; 3Department of Paediatric Nephrology, Alder Hey Children’s NHS Foundation Trust Hospital, Liverpool L14 5AB, UK

**Keywords:** IgA vasculitis, Henoch–Schonlein purpura, IgAV, HSP, nephritis, children, urine, biomarker

## Abstract

Chronic kidney disease is a recognised complication of immunoglobulin A vasculitis, (IgAV; formerly Henoch–Schonlein purpura—HSP). The pathophysiology of IgAV and why some patients develop significant renal involvement remains largely unknown. Identifying urinary inflammatory markers could direct targets for earlier intervention. The aim of this cross-sectional exploratory study was to perform a large protein array analysis to identify urinary markers to provide insight into the mechanisms of kidney inflammation in children with established IgAV nephritis (IgAVN). Determination of the relative levels of 124 key proteins was performed using commercially available proteome profiler array kits. Twelve children were recruited: IgAVN, *n* = 4; IgAV without nephritis (IgAVwoN), *n* = 4; healthy controls (HCs), *n* = 4. The urinary concentrations of twenty proteins were significantly different in IgAVN compared to IgAVwoN. The largest fold changes were reported for B-cell activating factor (BAFF), Cripto-1, sex-hormone-binding globulin and angiotensinogen. The urinary levels of complement components C5/C5a and factor D were also significantly elevated in patients with IgAVN. A total of 69 urinary proteins significantly raised levels in comparisons made between IgAVN vs. HCs and nine proteins in IgAVwoN vs. HCs, respectively. This study identified key urinary proteins potentially involved in IgAVN providing new insight into the pathophysiology. Further longitudinal studies with larger cohorts are needed to quantitatively analyse these biomarkers.

## 1. Introduction

Immunoglobulin A vasculitis (IgAV; formerly Henoch–Schonlein purpura—HSP [1]) is an immune-mediated vasculitis affecting small blood vessels. It is the most common form of childhood vasculitis with an estimated annual incidence of 20.4 per 100,000 children and a peak age of onset at approximately 6 years old [2]. It usually presents with a purpuric non-blanching rash with lower limb predominance, and it is commonly associated with abdominal pain, arthralgia/arthritis and renal involvement [3]. The disease typically self-resolves over a period of weeks to months in over 90% of children [4]; hence, treatment is mainly supportive [5]. Renal involvement (i.e., IgAV nephritis and IgAVN) is the most serious long-term manifestation of IgAV. IgAVN occurs in 40 to 50% of children, and it can present with a spectrum of manifestations that include microscopic haematuria and/or proteinuria, nephritic/nephrotic syndrome [4,6] and, rarely, rapidly progressive glomerulonephritis [7]. Although renal involvement is mainly self-limiting, 1–2% of children progress to chronic kidney disease (CKD) stage 5 [8]. For this reason, common clinical practice is that all children with IgAV should have at least 6 months of urinalysis testing and blood pressure monitoring following the first presentation [9,10]. The rising presence of proteinuria is suggestive of evolving nephritis, and the gold standard for confirming and grading the diagnosis is to perform a renal biopsy [5].

Histologically, IgAVN is characterised by mesangial deposition of pathogenic galactose-deficient IgA1 containing immune complexes, complement activation and neutrophil infiltrates in the glomerulus [11]. Yet, the exact pathophysiology of IgAV and the reasons why some patients develop significant renal involvement remains largely unknown. Identifying the variations in urinary proteome profiles between patients with and without nephritis may aid understanding of the pathophysiology of IgAVN and direct the identification of new treatment targets for future validation studies.

Urine offers an ideal biofluid to assess the presence of renal disease and to uncover potential biomarkers that may be more indicative of IgAVN and help elucidate mechanisms of disease. A recent systematic review of the existing literature highlighted that kidney injury molecule-1 (KIM-1), monocyte chemotactic protein-1 (MCP-1), N-acetyl-β-glucosaminidase (NAG) and urinary angiotensinogen (UAGT) seemed to be the most promising biomarkers in identifying the presence and/or severity of IgAVN [12]. The aim of this study was to perform a large exploratory protein array analysis of urinary markers of kidney inflammation and/or injury to provide insight into the pathophysiology of IgAV using a select cohort of children with established IgAVN. 

## 2. Materials and Methods

### 2.1. Definitions and Patient Selection

Children were recruited as part of the IgA Vasculitis Study, a single-centre observational longitudinal study at Alder Hey Children’s Hospital, Liverpool, UK, between 28 August 2019 and 13 October 2021. For this study, clinical data (i.e., history, medication, blood pressure, urinalysis, blood tests, histology reports, imaging report and disease progression) were collected at presentation and during follow up together with biosamples including urine.

Children of any sex aged <18 years old at diagnosis were eligible to take part in this study. Patients had to be diagnosed clinically with IgA vasculitis according to the EULAR/PRINTO/PRES criteria [3]. A minimum follow-up period of 12 months from sample collection was required, and a urine sample had to be obtained during this follow-up period. Patients were considered to have completed the 12 months follow up if they were discharged from the 6 month nurse-led HSP pathway [10] and did not have any further IgAV-related presentations to the institution. Exclusion criteria were as follows: (1) diagnosis of IgAV uncertain or in doubt; (2) other concurrent inflammatory or renal condition; (3) incomplete follow up; (4) urine sample obtained at a point greater than 12 months from diagnosis; (5) undergoing dialysis.

The patients with IgAV were subdivided according to the presence of nephritis. Nephritis (termed the “IgAVN” group) was defined as a urinary albumin to creatinine (UACR) > 30 mg/mmol at the time of sampling [3] and a kidney biopsy demonstrating IgAV prior to or within 7 days of sample collection. In order to identify cases with more severe renal inflammation, patients were selected from those with the highest UACR from the IgA Vasculitis study cohort. Patients with no renal involvement (termed the “IgAVwoN” group) were patients with a negative urine dipstick for protein and blood at the time of sampling and with an uncomplicated disease course, defined as no further presentation due to the fact of IgAV, a UACR < 10 mg/mmol at any point during their follow up and those who received no treatment apart from simple analgesia. A UACR of 0 mg/mmol was assumed for urine dipsticks negative for protein. Hypertension was defined as a systolic blood pressure above the 95th centile for the child’s age, sex and height for <16 years old or >140 mmHg for children 16 years and older [13]. 

Healthy controls were recruited to provide a one-off, age and sex-matched clean-catch, midstream spot urine sample. These were children aged <18 years old who were attending for noninflammatory day case investigations or surgery and with not known relevant medical history or not taking any regular medication.

### 2.2. Sample Processing

Healthy control urine samples were tested for bacterial contamination using urine dipstick testing, and they were discarded if they demonstrated positivity for leukocytes, nitrites, blood or >+1 for protein. Urine samples were centrifuged at 300× *g* for 10 min, transferred into a new falcon tube, centrifuged again at 300× *g* for 10 min and aliquoted into 1 mL sterile Eppendorf tubes for storage. Samples were stored at −80 °C and were thawed at room temperature and centrifuged for 5 min at 300× *g* on the day of the experiment.

### 2.3. Membrane Antibody Arrays

Human Kidney Biomarker Array Kit and Human XL Cytokine Array Kit (R&D Systems, Minneapolis, MN, USA) were used to assess the relative concentrations of 38 predefined markers of kidney damage and 105 inflammatory cytokines (total of 124 proteins, 19 were duplicated as they were evaluated in both kits). The full list of the proteins assessed can be found in Appendix A. Both kits were performed according to the manufacturer’s instructions. Briefly, each array kit consists of 4 nitrocellulose membranes with spotted capture and controls antibodies embedded in duplicates, each membrane allowing for 1 urine sample to be assessed (therefore 4 patients per kit). Three sets of each kit (12 membranes) were available, which allowed for the analysis of 12 different patients’ urine samples. The assays were conducted by performing 4 membranes at a time per day and subjects were intentionally divided to include at least one of each group in order to reduce selection bias. Due to the semi-quantitative nature of the membranes, we did not assess for intra/inter-variability, but we calculated Pearson’s correlation coefficient on the fold changes of the duplicated proteins.

For clarity, the Human Kidney Biomarker Array kit will be further referred to as “Kit K” and the Human XL Cytokine Array kit as “Kit C” throughout the study.

#### 2.3.1. Human Kidney Biomarker Array—Kit K

The nitrocellulose membranes were blocked for 1 h using the provided array buffer and incubated with the capture antibody cocktail and 500 µL of thawed urine at 4 °C overnight on a rocking platform shaker. Urine samples were frozen again pending Kit C analysis. Membranes were then washed, incubated with horseradish peroxidase-conjugated Streptavidin (Streptavidin-HRP) and chemiluminescence reagents. Signal intensity was measured using enhanced chemiluminescence on the ChemiDoc MP Imaging System (Bio-Rad Laboratories Ltd., Watford, UK). 

#### 2.3.2. Human XL Cytokine Array Kit—Kit C

Urine samples were thawed and centrifuged at 300× *g* for 5 min. Membranes were blocked for 1 h using the provided array buffer and incubated with 400–450 µL of urine at 4 °C overnight on a rocking platform shaker. Membranes where then washed, incubated with biotinylated detection antibodies, followed by the addition of Streptavidin-HRP and chemiluminescence reagents. Signal intensity was measured using enhanced chemiluminescence on the ChemiDoc MP Imaging System (Bio-Rad Laboratories Ltd., Watford, UK).

#### 2.3.3. Creatinine Quantification

All results were corrected for the urinary creatinine concentration and automated quantification of urinary creatinine was run by the Biochemistry Department (Alder Hey Children’s NHS Foundation Trust, Liverpool, UK) using an enzymatic creatinine method [14] with an Alinity Ci System analyser (Abbott, Abbott Park, IL, USA). 

### 2.4. Ethical Approval

All procedures involving human subjects were conducted in accordance with NIHR Good Clinical Practice, HTA Codes of Practice, the Declaration of Helsinki and comparable ethical standards. This study was part of the IgA Vasculitis study which was approved by HRA and Health and Care Research Wales (HCRW) on 21 June 2019 (REC 17/NE/0390, protocol UoL001347, IRAS 236599). Written informed consent was obtained from parents and children prior to any study-related procedure.

### 2.5. Data Analysis

The intensity of each individual chemiluminescence signal was measured using ImageJ software (NIH) [15] and ImageJ Protein Array Analyser Macro [16] with automatic background removal. To determine the relative concentration of each protein, the average value of the duplicates was measured and normalised to the average of the 6 positive control points, the volume of urine aliquot used and the urinary creatinine concentration. For each protein, fold change was obtained by calculating the ratios of the average relative concentration per group for IgAVN/IgAVwoN, IgAVN/HC and IgAVwoN/HC. Data are presented as the fold change (FCx) between two groups.

### 2.6. Statistical Analysis

Clinical and demographical data were compared with the Statistical Package for the Social Science (SPSS) version 27.0 software for Windows (IBM Corp, Armonk, NY, USA). Due to the small sample size, data were assumed to be non-normally distributed and Mann–Whitney U/Kruskal–Wallis with Dunn–Bonferroni post hoc tests were used for continuous variables. Pearson’s chi-square was applied to categorical variables. For the analytes, multivariate principal component analysis (PCA) was performed using the MetaboAnalyst 5.0 online platform [17] with log2 transformation and pareto scaling. The Student’s *t*-test was applied on the log2 transformed data. *p*-Values and fold change (FC) were combined using the “EnhancedVolcano” package [18] in R Statistical Software version 4.1.2 (R Foundation for Statistical Computing, Vienna, Austria) to generate volcano plots and identify significant proteins. A *p*-value of <0.05 was considered statistically significant. Protein concentrations were considered to be elevated for FC ≥ 2.0 (2x) and reduced for FC ≤ 0.5 (0.5x).

## 3. Results

### 3.1. Paediatric Cohort

The study identified 8 children with IgAV from a cohort of 51 recruits to be included in this exploratory analysis. This included four children with nephritis selected as detailed previously and four children without any evidence of past or recent nephritis (termed as 4 IgAVN and 4 IgAVwoN). Four HCs were randomly selected from a cohort of 18 HC children who had been previously recruited to the IgA Vasculitis study. Demographics and baseline characteristics are presented in Table 1. The male:female ratio was 1:1, and the median age was 7.6 years old (range: 4.0–13.4). In those patients with IgAVN, renal involvement was confirmed histologically in all patients (grade II, *n* = 1; grade IIIb, *n* = 3), and the median UACR at the time of sample collection was 542.2 (110.4–2357.7) mg/mmol. All patients within the IgAVN cohort had nephrotic range proteinuria as an indication for conducting a renal biopsy with a mildly reduced serum albumin concentration but no evidence of peripheral oedema. They all had normal blood pressure, normal serum complement titres and normal renal function. The histological analysis demonstrated diffuse proliferative glomerulonephritis with <50% crescents in three patients and focal proliferative glomerulonephritis in one patient. All patients had IgA-positive staining on immunofluorescence (IF) and three out of the four patients had C3 positivity on IF. One patient had patchy evidence of acute tubular inflammation and one patient had <5% interstitial fibrosis. Regarding treatment at the time of sample collection, one IgAVN patient was taking oral prednisolone and one patient was taking both prednisolone and azathioprine. Subsequent treatments included azathioprine and prednisolone for one patient; mycophenolate moefetil, ACEi and prednisolone for another; ACEi only for the third patient; prednisolone and azathioprine for the fourth patient. One healthy control was prescribed somatotropin at the time of sample collection. None of the IgAVwoN patients received any medication. At 12 months, all of the IgAVwoN patients were discharged and three patients with IgAVN remained under follow up.

### 3.2. Exploratory Data Analysis

Principal component (PC) analysis was conducted to visualise the data sets using the first two principal components, and the score plots for each kit are presented in Figure 1a,b (Kit K: PC1 × PC2 = 82.8%; Kit C: PC1 × PC2 = 82.7%). The models demonstrated clear separation between IgAVN and both the IgAVwoN and HC groups. The IgAVN and IgAVwoN clusters demonstrated some areas of overlap in Kit C. 

### 3.3. Comparison between IgAVN and IgAVwoN

Urinary levels of 20 proteins (5 in Kit K: 3.1% and 15 in Kit C: 14.3%) were significantly increased in IgAVN compared to IgAVwoN (Figure 2a,b). Proteins significant in either IgAVN vs. IgAVwoN or IgAVN vs. HC comparisons are shown in Table 2; full results can be found in Appendix A. The main features identified were B-cell activating factor (BAFF; 9.7x), Cripto-1 (7.8x), sex-hormone-binding globulin (SHBG; 7.6x), angiotensinogen (AGT; 6.5x), apolipoprotein A1 (ApoA1; 6.0x) and epithelial growth factor receptor (EGF-R; 5.8x). One interleukin (i.e., IL-5) and two components of the complement system (i.e., C5/C5a and complement factor D—(CFD)) were also significant, with a fold change of, respectively, 2.0, 4.6 and 2.0. Endoglin (1.75x; *p* = 0.021) and HGF (1.80x; *p* = 0.028) were both statistically significant but did not reach the fold-change criteria. The urinary concentration of one marker was reduced, but this was not statistically significant (myeloperoxidase; 0.4x; *p* = 0.345).

### 3.4. Comparison between IgAVN and the HCs

Analysis of IgAVN versus the HCs revealed 68 proteins with significantly increased urinary concentrations (Figure 3a,b and Table 2; full results in Appendix A). The highest reported fold change was BAFF (26.5x) followed by angiotensinogen (22.6x), SHBG (13.3x) and MCP-1 (12.0x). Eighteen interleukins out of the 26 assessed by the arrays were significantly elevated compared to the control levels. All of the significant proteins identified by the IgAVN vs. IgAVwoN comparison were also significant in this comparison. No proteins had reduced levels between the two groups.

### 3.5. Comparison between IgAVwoN and the HCs

The concentrations of nine proteins were increased in the urine of IgAVwoN patients when compared to the HCs (Figure 4a,b). MCP-1 levels were statistically significantly increased (3.5x; *p* = 0.001); angiotensinogen was increased, although not reaching statistical significance (3.5x; *p* = 0.055). Urinary IL-13 (2.8x; *p* = 0.049), leptin (2.6x; *p* = 0.022), IL-19 (2.5x; *p* = 0.029), GH (2.4x; *p* = 0.017), HGF (2.4x; *p* = 0.020), IL-4 (2.3x; *p* = 0.036), IL-5 (2.3x; *p* = 0.016) and CXCL1 (2.1x; *p* = 0.017) concentrations were all elevated in IgAVwoN patients compared to the HCs.

### 3.6. Correlation between the Kit K and C

A total of 19 proteins were assessed in both Kit K and Kit C. Moderate positive correlation [19] (Pearson’s correlation coefficient of 0.640) was found between the two kits when comparing the reported average fold changes. MCP-1 reached statistical significance in all comparisons in Kit K as well as in IgAVN vs. IgAVwoN in Kit C and demonstrated a very similar *p*-value, although it was located on either side of the cut-off, for IgAVN vs. IgAVwoN (*p* = 0.061) in Kit C. CXCL1 was found to be significant in all comparisons in Kit C only. The fold changes and *p*-values of six proteins that were reported as statistically significant in at least one kit are reported in Table 3. There was agreement on a further 13 proteins which did not reach statistical significance in both kits.

## 4. Discussion

This study aimed to use an exploratory small cohort to assess the relative concentrations of 124 known urine markers of kidney inflammation and injury in children with IgAV to identify differences between those with and without established renal involvement to provide insight into pathophysiological mechanisms. The demographic data, renal parameters and histological features of our cohort were typical of how the majority of children present with IgAVN, and we report increased relative urinary levels of 20 proteins in IgAVN when compared to IgAVwoN. To our knowledge, this exploratory study is the first to evaluate such an extensive panel of urinary proteins in children with IgAVN. Although the study cohort was limited by size, we report some novel findings that provide new insight into the pathophysiology of IgAVN. 

A recognised feature of IgA-related renal diseases is the presence of galactose deficient IgA, known as Gd-IgA1, and glycan-specific IgG and IgA antibodies, which are found at increased circulating levels in IgAVN [11]. They are mainly derived from activated B-cell lymphocytes, and effective B-cell maturation relies upon B-cell activating factor (BAFF) [20]. In our study, urinary BAFF levels increased by 9.7-fold between IgAVN and IgAVwoN and by 26.5-fold between IgAVN and HCs. The role of BAFF in IgAV remains unclear, as it has only been explored in one study that did not find any association between BAFF gene polymorphisms and IgAV [21]. BAFF is a cytokine of the tumour necrosis factor (TNF) family, and some evidence suggests it could activate the tumour necrosis receptor-associated factor 6 (TRAF6)/NF-kB pathway in glomerular mesangial cells, consequently contributing to the pathogenesis of IgA nephropathy [22]. Vincent and colleagues were unable to detect urinary BAFF in adult patients with IgA nephropathy [23]; however, in mice, overexpression of BAFF resulted in an IgA nephropathy–renal phenotype and hyper-IgA syndrome [24,25]. In addition to IgA nephropathy, raised serum BAFF levels have also been linked to other autoimmune diseases such as systemic lupus erythematosus (SLE) and multiple sclerosis [20]. Increased urinary BAFF levels and its usefulness as a biomarker has been reported in lupus nephritis [23,26]. Our data suggest that BAFF could be involved in the pathophysiology of IgAVN, and this warrants further analysis in validation studies.

One of the other significantly elevated proteins in this study was Cripto-1, an embryonic protein that is re-expressed during inflammation, wound healing and tumorigenesis [27]. Several in vitro studies have shown that Cripto-1 modulates macrophage cytokine secretion, resulting in elevated concentrations of pro-inflammatory cytokines TNF-α, IL-6, IL-1β and anti-inflammatory cytokine IL-10 [27,28], all of which have also been linked to the pathogenesis of IgAV and IgAVN [29]. In addition, overexpression of Cripto-1 has been reported in other diseases, including Crohn’s disease, ulcerative colitis and rheumatoid arthritis [27], suggesting a link to autoinflammatory processes. Increased urinary concentrations of Cripto-1 in our IgAVN cohort supports a potential role of this modulating protein in IgAVN.

The potential biological significance of our finding of increased urinary sex-hormone-binding globulin (SHBG) levels in IgAVN is unclear. SHBG is a glycoprotein primarily synthesised by the liver, and it is involved in transporting androgen and oestrogen. SHBG serum levels are stable in childhood and decrease at puberty, especially in boys [30]. In vitro, SHBG suppresses inflammation in macrophages [31], and it is downregulated by TNF-α [32] and IL-1β [33]. In our cohort, two of the IgAVN patients were taking prednisolone, and the pubertal status of our cohort was not documented. The impact of synthetic glucocorticoids on urinary SHBH is unknown, although some studies have reported lowered SHBG serum levels for patients undergoing treatment with corticosteroids [34]. 

The significantly increased concentration of urinary angiotensinogen in this study is not too surprising, because extensive evidence supports the existence of an intrarenal RAAS system that contributes to kidney disease progression [35], and the benefits of RAAS inhibition, as a renoprotective treatment, is well recognised for glomerular diseases [36]. Furthermore, the RAAS system may have a direct role in complement pathway activation through renin-mediated C3 cleavage as demonstrated in vitro and suggested through a study evaluating the efficacy of direct renin inhibition (aliskiren) in dense deposit disease where reduced systemic and renal complement activation was observed [37]. ACE gene polymorphisms have also been associated with IgAV susceptibility [38]. The majority of the circulating AGT is produced by the liver but local renal production by the proximal tubules has been demonstrated in animal models [39]. In a study by Nishiyama and colleagues, urinary AGT (UAGT) levels were raised in patients with IgA nephropathy, and this was shown to correlate with renal tissue gene expression of AGT and angiotensin II immunoreactivity. Thus, UAGT was proposed to provide a specific index of intrarenal RAAS activity in patients with IgA nephropathy [40]. We found that urinary UAGT concentrations increased by 6.5-fold in IgAVN compared to IgAVwoN, and this was observed by two previous studies that found remarkably elevated UAGT levels in children with IgAVN [41,42]. However, urinary renin levels were not significantly different across the groups in the current study. Interestingly, in our cohort, UAGT levels were also increased in patients without nephritis, although this finding was not statistically significant (*p* = 0.055). This could be due to the fact of a spurious finding, but other possible explanations include the relationship of the RAAS system with systemic vascular inflammation [43] and/or the presence of subclinical kidney inflammation not detectable using current methods of disease activity monitoring.

We report elevated levels of urinary apolipoprotein A1 (ApoA1), which is a novel finding in IgAVN. ApoA1 is the main protein component of high-density lipoproteins (HDL) [44], and although HDL has a protective role in cardiovascular disease and it has been widely studied, more recent evidence suggests HDL may contribute to kidney disease [45] and autoimmunity [46]. A recent study reported elevated urinary ApoA1 in children with various kidney diseases (including 18 children with glomerulonephritis) [47], whilst children with nephrotic syndrome in remission, nephrolithiasis, polycystic kidney disease, transplant or hypertension did not show higher ApoA1 levels [47]. Identifying whether ApoA1 is excreted in a modified form and its effect on downstream lipid pathways could provide more information on the renal handling of lipoproteins and their role in this disease.

Activation of both the alternative and mannose-binding lectin complement pathways have been demonstrated in IgAV [11], with skin and mesangial deposits reported to contain complement components C3 and C5b-9 [48,49]. In addition, C4d and C5b-9 positivity in renal histology has been associated with poor renal outcomes in patients with IgAVN and IgA nephropathy [50]. The findings of increased C5/C5a and complement factor D concentrations in the urine of children with IgAVN compared to IgAVwoN (fold changes of, respectively, 4.6 and 2.6) raises the potential role of the complement system in the pathophysiology of this condition and supports this pathway as a possible treatment target. C5 cleaves into C5a—a potent neutrophil chemoattractant—and C5b, which ultimately leads to the formation of the membrane attack complex (MAC) [51]. C5a has also been shown to increase production of IL-8, MCP-1 and ICAM-1 by endothelial cells in vitro [48], and these markers were also increased in our cohort. Further confirmation regarding the role of the complement pathway in IgAVN is warranted. 

Our findings are consistent with those reported in a recent systematic literature review that identified urinary biomarkers in children with IgAVN [12]. The current summary of the literature supported the significance of UAGT, β2-microglobulin, KIM-1 and MCP-1 [12]. In addition, individual studies have reported elevated urinary levels of MIF [52], MMP-9 [53], IL-6, IL-8 and IL-10 [54] in paediatric IgAVN. Our study did not find elevated levels of β2-microglobulin, MMP-9 and IL-6/8/10 in IgAVN; however, it is important to note that this was a small preliminary study using a semi-quantitative technique. Dyga and colleagues did not find any significant association between the concentrations of urinary NGAL and L-FABP [55], which is in keeping with our findings. Our findings of elevated urinary proteins, which could result from tubulointerstitial inflammation/injury (KIM-1 [56], ApoA1 [47,57], EGF-R [58], AGT [59]), further support that IgAVN includes some degree of tubular inflammation in addition to the glomerulonephritis as previously proposed by Williams et al. [12]. Finally, it remains unclear whether these markers are specific to IgAVN or result from pathophysiological features shared between different renal diseases, such as excessive inflammation, or a by-product of proteinuria. While this study was limited in terms of sample size, it provides exploratory data to direct future studies, and it demonstrates that urinary protein profiles do vary between IgAV and IgAVN patients. Future work will look to utilise more robust quantitative methods, such as ELISAs and LC-MS/MS, to validate the findings of the most highly expressed proteins using a larger patient cohort.

## 5. Conclusions

The current study utilised a small number of patients in a cross-sectional design to explore a large number of urinary proteins in patients with established IgAVN. It was able to uncover some interesting variations in the urinary protein profiles that may help elucidate further mechanisms of pathophysiology and represent potential drug targets for intervention. However, there are limitations to this study, predominantly related to the sample size, and these findings require validation using larger cross-sectional cohorts. Furthermore, longitudinal evaluation would allow for the evolution of nephritis to be better characterised, which would significantly improve our mechanistic understanding of the development of IgAVN. 

## Figures and Tables

**Figure 1 children-09-00622-f001:**
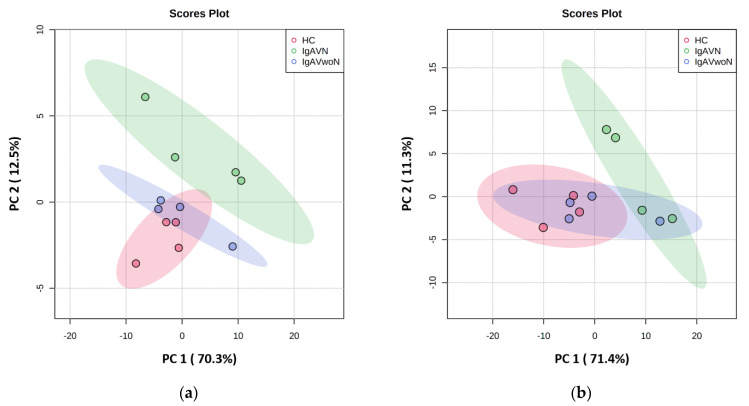
Exploratory data analysis representation: (**a**) Kit K: scores plot between principal component (PC) 1 and PC2 with the explained variance shown in brackets; (**b**) Kit C: scores plot between PC1 and PC2 with the explained variance shown in brackets.

**Figure 2 children-09-00622-f002:**
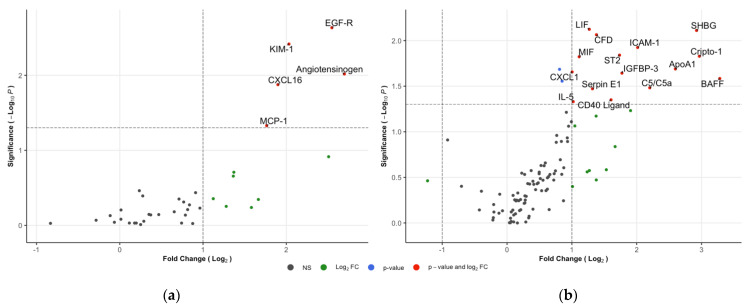
Volcano plots of IgAVN vs. IgAVwoN: (**a**) Kit K; (**b**) Kit C. The horizontal lines represent the statistical significance cut off (−log10(0.05)). Vertical lines represent the fold-change cut-off (log2(0.5)−log2(2.0)).

**Figure 3 children-09-00622-f003:**
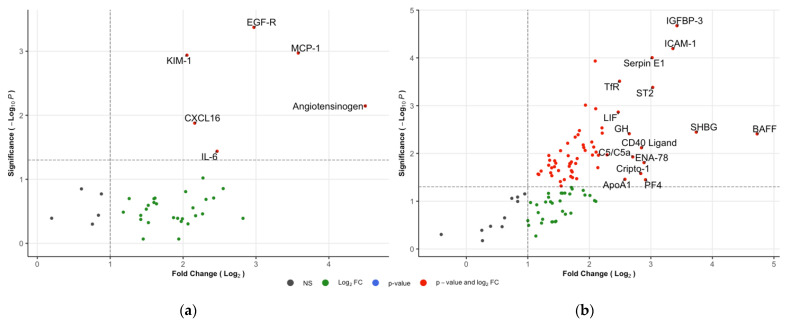
Volcano plots of IgAVN vs. IgAVwoN: (**a**) Kit K; (**b**) Kit C. The horizontal lines represent the statistical significance cut off (−log10(0.05)). Vertical lines represent the fold-change cut-off (log2(2.0)). Fifteen top hits in term of FC are labelled in (**b**).

**Figure 4 children-09-00622-f004:**
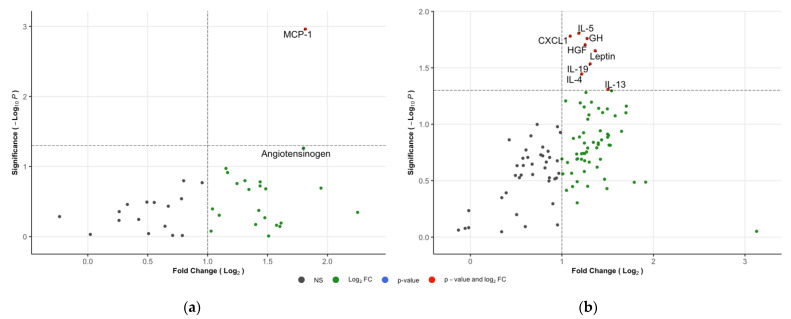
Volcano plots of IgAVwoN vs. HCs: (**a**) Kit K; (**b**) Kit C. The horizontal lines represent the statistical significance cut-off (−log10(0.05)). Vertical lines represent the fold-change cut-off (log2(2.0)).

**Table 1 children-09-00622-t001:** Patients characteristics at baseline.

	Overall	IgAVN	IgAVwoN	HC	*p*-Value
*n*	12	4	4	4	-
Male/female	6/6	2/2	2/2	2/2	1.000
Age, years ^a^	7.6 [4.0–13.4]	9.9 [5.2–12.2]	5.1 [4.0–6.8]	8.8 [7.5–13.4]	0.058
Time from diagnosis to sample (weeks) ^a^	14.8 [1.0–32.0]	14.8 [5.3–25.3]	19.2 [1.0–32.0]	-	0.773
Hypertension ^b^	2	2	0	-	0.102
Serum creatinine, mg/dL ^a^	42.0 [32.0–54.0]	43.5 [33.0–54.0]	32.0 [32.0–32.0] ^c^	-	0.400
Urinary creatinine, mmol/L ^a^	7.8 [1.2–19.9]	6.6 [1.6–13.2]	7.8 [1.2–9.4] ± 3.6	9.3 [6.8–19.9]	0.668
UACR, mg/mmol ^a,d^	0.0 [0.0–2357.7]	542.2 [110.4–2357.7]	0.0 [0.0–0.0]	0.0 [0.0–0.0]	-
Biopsy proven nephritis ^b^	-	4	0	-	-
ISKDC Grade II ^b^ IIIb ^b^	- -	1 3	- -	- -	- -
Medications ^b^	3	2	0	1	-
Corticosteroids ^b^	2	Prednisolone (2)	-	-	-
DMARDs ^b^	1	Azathioprine (1)	-	-	-
Other ^b^	1	-	-	Somatropin (1)	-
Follow up at 12 months					
Discharged	5	1	4	-	
Still under follow up	3	3	-	-	

^a^ Median [range]; ^b^
*n*; ^c^ serum creatinine only available for one IgAVwoN patient; ^d^ UACR of 0 mg/mmol assumed if urine dipstick negative for protein. UACR: urinary-albumin-to-creatinine ratio; ISKDC: International Study for Kidney Disease in Children Classification; DMARDs: disease modifying antirheumatic drugs.

**Table 2 children-09-00622-t002:** A summary of the urinary proteins that were statistically significantly changed in both assays combined as sorted in order of descending fold change (IgAVN vs. IgAVwoN). Bold text represents duplicated proteins that were measured in both kits where results from Kit C are reported. * *p* < 0.05.

Protein	IgAVN vs. IgAVwoN		IgAVN vs. HCs	
	Fold Change	*p*-Value	Fold Change	*p*-Value
BAFF	9.7	0.026 *	26.5	0.004 *
Cripto-1	7.8	0.015 *	7.1	0.026 *
SHBG	7.6	0.008 *	13.3	0.004 *
Angiotensinogen	6.5	0.010 *	22.6	0.007 *
ApoA1	6.0	0.020 *	6.0	0.035 *
EGF-R	5.8	0.002 *	7.8	<0.001 *
C5/C5a	4.6	0.033 *	6.5	0.012 *
KIM-1	4.1	0.004 *	4.1	0.001 *
ICAM-1	4.0	0.012 *	10.2	<0.001 *
ENA-78	3.7	0.059	7.4	0.016 *
CXCL16	3.7	0.013 *	4.5	0.013 *
IGFBP-3	3.4	0.023 *	10.7	<0.001 *
MCP-1	3.4	0.047 *	12.0	0.001 *
ST2	3.3	0.014 *	8.2	<0.001 *
CD40 Ligand	3.0	0.045 *	7.2	0.008 *
PF4	2.9	0.261	7.5	0.036 *
CFD	2.6	0.009 *	3.4	0.005 *
GH	2.6	0.067	6.3	0.004 *
**IL-6**	2.6	0.196	5.5	0.036 *
Serpin E1	2.5	0.034 *	8.1	<0.001 *
LIF	2.4	0.008 *	5.5	0.001 *
MIF	2.2	0.015 *	3.1	0.006 *
Fas Ligand	2.1	0.086	3.8	0.008 *
IL-5	2.0	0.047 *	4.6	0.003 *
**CXCL1**	2.0	0.022 *	4.3	<0.001 *
IL-4	2.0	0.078	4.6	0.004 *
TfR	1.9	0.087	5.6	<0.001 *
RANTES	1.9	0.128	4.4	0.011 *
IL-8	1.9	0.117	2.9	0.039 *
**MCP-1**	1.9	0.061	4.1	0.006 *
IL-22	1.8	0.248	4.4	0.020 *
HGF	1.8	0.028 *	4.3	0.001 *
IP-10	1.8	0.128	2.8	0.023 *
IGFBP-2	1.8	0.203	4.9	0.011 *
Endoglin	1.8	0.021 *	2.7	0.014 *
MIG	1.7	0.110	3.8	0.001 *
Flt-3 Ligand	1.7	0.131	2.7	0.019 *
Leptin	1.7	0.268	4.3	0.009 *
**IL-10**	1.7	0.288	3.3	0.032 *
IL-19	1.5	0.320	3.8	0.009 *
G-CSF	1.5	0.220	3.6	0.003 *
PDGF-AA/BB	1.5	0.236	4.2	0.007 *
TARC	1.5	0.340	4.1	0.011 *
GM-CSF	1.4	0.234	3.2	0.016 *
Angiopoietin-1	1.4	0.326	3.5	0.034 *
MIP-1α/MIP-1β	1.4	0.277	2.6	0.014 *
IL-15	1.4	0.289	3.2	0.011 *
IL-33	1.4	0.363	3.7	0.007 *
IL-31	1.4	0.371	3.3	0.015 *
IL-23	1.4	0.431	3.3	0.024 *
MCP-3	1.3	0.377	3.2	0.016 *
MIP-3α	1.3	0.349	2.5	0.018 *
IL-16	1.3	0.374	2.7	0.016 *
I-TAC	1.2	0.370	2.9	0.009 *
Relaxin-2	1.2	0.510	2.9	0.048 *
IL-24	1.2	0.504	3.2	0.019 *
IL-13	1.2	0.603	3.4	0.016 *
IFN-γ	1.2	0.609	3.2	0.030 *
IL-34	1.2	0.554	2.6	0.025 *
Pentraxin-3	1.1	0.732	3.0	0.035 *
IL-27	1.1	0.570	3.3	0.023 *
**TNF-α**	1.1	0.469	2.3	0.023 *
Dkk-1	1.1	0.562	2.5	0.011 *
IL-32	1.1	0.558	3.5	0.004 *
BDNF	1.0	0.757	2.6	0.020 *
PDGF-AA	1.0	0.735	2.2	0.027 *
MIP-3b	0.9	0.484	3.5	0.013 *
**Thrombospondin-1**	0.9	0.781	2.6	0.029 *
TGF-α	0.9	0.929	2.3	0.028 *

**Table 3 children-09-00622-t003:** Proteins reported significant by one comparison in at least one kit. Data are presented as the fold change (*p*-value). The bold text highlights discrepancies in the results of significance between the kits.

Protein	IgAVN vs. IgAVwoN		IgAVN vs. HC		IgAVwoN vs. HC	
	Kit K	Kit C	Kit K	Kit C	Kit K	Kit C
CXCL1	**1.8 (0.534)**	**2.0 (0.022)**	**5.0 (0.206)**	**4.3 (<0.001)**	**2.8 (0.539)**	**2.1 (0.017)**
IL-6	2.6 (0.196)	1.8 (0.293)	5.5 (0.036)	3.3 (0.052)	2.1 (0.496)	1.8 (0.197)
IL-10	1.6 (0.659)	1.7 (0.288)	**4.8 (0.346)**	**3.3 (0.032)**	3.1 (0.642)	2.0 (0.203)
MCP-1	3.4 (0.047)	1.9 (0.061)	12.0 (0.001)	4.1 (0.006)	**3.5 (0.001)**	**2.2 (0.134)**
Thrombospondin-1	1.4 (0.719)	0.9 (0.781)	**3.7 (0.397)**	**2.6 (0.029)**	2.6 (0.673)	2.9 (0.153)
TNF-α	1.6 (0.446)	1.1 (0.469)	**2.7 (0.366)**	**2.3 (0.023)**	1.6 (0.963)	2.1 (0.385)

## Data Availability

The data presented in this study are available in the Appendix A.

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
