# Peer review of "Urinary Protein Array Analysis to Identify Key Inflammatory Markers in Children with IgA Vasculitis Nephritis"

_children, 2022, doi:10.3390/children9050622_

Round 1

Reviewer 1 Report

The aim of this study was to perform a large protein array analysis of urinary markers of kidney inflammation that may distinguish children with IgAVN.It is this study is interesting .It has certain reference value for clinical work.But the sample is small. 

Author Response

Please find attached PDF file summarising response to all reviewers 

Reviewer 2 Report

The topic is interesting. IgAVN is a relatively benign disease in children, but long-term cohort sudies have revealed a proportion of patients with persistent renal dysfunction.

The main flow of this observational study is small number of patients. Introduction should be more focused and shortened. 

Materials and Methods - please state which investigations, instead of "other" (L75-76)

Results - Section 3.1: number of patients, site, study period - to Materials (the name of the hospital is already written) 

Do not begin the sentence with number (12 children....)

Tables 2 and 3 could be combined in one

There are several ERROR! Reference source not found! throughout the text (ie. line 186 - 211 - 212 ...

Reviewer 3 Report

This manuscript is a study that attempts to elucidate the pathophysiology of IgA vasculitis nephritis by comparing the urinary inflammatory markers of IgA vasculitis nephritis and IgA vasculitis without nephritis.  I read your manuscript with great interest. 

Unfortunately, the number of cases is very small.  I would like authors to discuss and announce the increase in the number of cases, including the differences from other renal diseases.

Round 2

Reviewer 2 Report

The main and serious flaw is still a small choort. I suggest to the authors to continue their interesting research and publish their results with the relevant number of cases.

Author Response

Point 1: small cohort size 

Thank you to reviewer 2 for their additional feedback. We have further improved the manuscript to emphasise the limitation in sample size throughout the text (abstract, introduction paragraph 2 and 3, discussion paragraph 1 and 2, conclusion) and the exploratory nature of the study (already present in abstract, introduction, added to discussion paragraph 1). Further additions include the emphasis on the need for larger, quantitative, validation studies as a follow up to this initial work which we agree is very important. We believe that these changes have further improved and clarified the intention of the manuscript.